# Growth deficiency and enhanced basal immunity in *Arabidopsis thaliana* mutants of *EDM2*, *EDM3* and *IBM2* are genetically interlinked

**Jianqiang Wang**¤, **Thomas Eulgem** *

Center for Plant Cell Biology, Department of Botany and Plant Sciences, Institute for Integrative Genome Biology, University of California, Riverside, Riverside, California, United States of America

¤ Current address: International Institutes of Medicine, The Fourth Affiliated Hospital of Zhejiang University, School of Medicine, Yiwu, Zhejiang, China

* thomas.eulgem@ucr.edu

**Data Availability Statement:** All relevant data are within the manuscript and its Supporting information files.

## Abstract

Mutants of the *Arabidopsis thaliana* genes, *EDM2 (Enhanced Downy Mildew 2)*, *EDM3 (Enhanced Downy Mildew 3)* and *IBM2 (Increase in Bonsai Methylation 2)* are known to show defects in a diverse set of defense and developmental processes. For example, they jointly exhibit enhanced levels of basal defense and stunted growth. Here we show that these two phenotypes are functionally connected by their dependency on the salicylic acid biosynthesis gene *SID2* and the basal defense regulatory gene *PAD4*. Stunted growth of *edm2*, *edm3* and *ibm2* plants is a consequence of up-regulated basal defense. Constitutively enhanced activity of reactive oxygen species-generating peroxidases, we observed in these mutants, appears also to contribute to both, their enhanced basal defense and their growth retardation phenotypes. Furthermore, we found the histone H3 demethylase gene *IBM1*, a direct regulatory target of EDM2, EDM3 and IBM2, to be at least partially required for the basal defense and growth-related effects observed in these mutants. We recently reported that *EDM2*, *EDM3* and *IBM2* coordinate basal immunity with the timing of the floral transition by gradually reducing the extent of this defense mechanism prior to flowering. Together with these observations, data presented here show that at least some of the diverse phenotypic effects in *edm2*, *edm3* and *ibm2* mutants are genetically interlinked and functionally connected. Our new results show that repression of basal immunity by *EDM2*, *EDM3* and *IBM2* limits negative impact on growth and development.

## Introduction

Sustained or constitutive activation of immune responses in plants is often associated with reduced growth and other developmental abnormalities, a phenomenon related to the concept of defense-growth trade-off. It has long been believed that such effects are due to competition of immunity-related and developmental processes for limited metabolic resources [1]. When

**Funding:** This work was supported by National Science Foundation (NSF) grant IOS-1457329 to TE. URL of funders web site: https://www.nsf.gov/div/index.jsp?div=IOS The funders had no role in study design, data collection and analysis, decision to publish, or preparation of the manuscript.

**Competing interests:** The authors have declared that no competing interests exist.

plants are attacked by microbial pathogens, metabolic resources are preferentially allocated to defense, resulting in enhanced immunity, but at the expense of reduced growth [2, 3].

The *Arabidopsis thaliana* (Arabidopsis) Plant Homeodomain (PHD) finger protein-encoding *EDM2* gene and the RNA Recognition Motif (RRM) domain protein-encoding genes *EDM3/AIPP1* (hereafter *EDM3*) and *IBM2/ASI1/SG1* (hereafter *IBM2*) are known to jointly affect multiple immunity-related and developmental processes [4–12]. They cooperate in a strong race-specific pathogen defense mechanism mediated by the disease resistance gene *RPP7*, that encodes an NLR-type immune receptor [13–15]. They also work together in an unspecific immune response, effective against a wide range of pathogens, termed basal immunity or basal resistance [16, 17]. While they promote *RPP7*-mediated immunity, *EDM2*, *EDM3* and *IBM2* suppress basal defense responses. The latter process is coordinated by these three genes with the floral transition [17]. Besides this and other developmental roles, *EDM2*, *EDM3* and *IBM2* promote overall growth, as their mutants exhibit reduced fresh weight and smaller rosette leaves [5, 11].

Mircoarray and RNA-seq studies with their mutants have shown *EDM2*, *EDM3* and *IBM2* to affect transcript levels of large overlapping sets of genes and transposable elements, while epigenome profiling revealed joint effects on methylation of the nucleobase cytosine and/or the repressive histone mark H3K9me2 at numerous chromatin sites [5, 8–10, 12, 16]. By chromatin-immunoprecipitation (ChIP) the EDM2, EDM3 and IBM2 proteins were found to co-localize at some of these loci [7–10, 12, 13, 16, 18]. *RPP7* and the histone H3K9 demethylase gene *IBM1* are among the direct target sites of EDM2, EDM3 and IBM2. Upon co-recruitment to heterochromatic stretches at each of these two (and several other) target loci, they prevent premature transcript termination at alternative polyadenylation signals, thereby promoting the synthesis of full-length mRNAs [8–10, 12, 13, 15, 16]. Consistent with their co-localization at chromatin sites, physical interactions among these three proteins were observed. While both EDM2 and IBM2 appear to directly interact with EDM3, interactions between EDM2 and IBM2 seem indirect and require EDM3 as a molecular bridge [12]. Recent results strongly suggested that EDM3/IBM2 interactions can be isoform specific [17]. Due to alternative splicing, two distinct protein isoforms are expressed for each of these two genes; a shorter one (EDM3S; IBM2S) and a longer one (EDM3L; IBM2L). Regarding their role in the suppression of basal immunity and the promotion of the floral transition, only the longer isoforms (EDM3L and IBM2L) are involved and not the shorter ones i.e. EDM3S and IBM2S [17].

Consistent with their role in suppressing the basal defense response, constitutive up-regulation in the transcripts of large overlapping sets of defense-related genes and immune receptor genes has been observed in the mutants of EDM2, EDM3 and IBM2 [15–17]. Thus, *edm2*, *edm3* and *ibm2* mutants exhibit three typical effects common to many other Arabidopsis mutants with constitutively elevated levels of basal immunity [19–23]: (1) constitutively elevated expression of defense genes, (2) enhanced basal defense and (3) retarded growth. However, it has been unclear, weather these effects are independent or causally connected.

Here we show that enhanced basal defense and growth retardation in *edm2*, *edm3* and *ibm2* mutants are genetically coupled by their dependency on the salicylic acid-associated defense genes *SID2* and *PAD4*, implying that the latter effect is likely a consequence of the former one. We further present data suggesting that increased peroxidase activity, as a part of the constitutive basal defense response, is a partial cause of reduced *edm2*, *edm3* or *ibm2* growth rates. In addition we found that the direct EDM2, EDM3 and IBM2 target gene *IBM1*, is partially required for the basal defense and growth-related phenotypes. Each of these effects is mediated by the longer isoforms of EDM3 and IBM2 but not the shorter ones. Together with our previous report on the coordination of basal defense with the timing of the floral transition by *EDM2*, *EDM3* and *IBM2*, our new results show that these three chromatin-associated

regulators link various defense and developmental processes. This is likely contributing to a balanced defense-growth trade-off that enables efficient developmental transitions and limits growth penalties due to energetically costly basal immune responses.

## Results

### *EDM2*, *EDM3L* and *IBM2L* positively affect plant growth

Previously, we showed that *edm2* mutants exhibit decreased fresh weight and rosette leaf expansion compared to their wild type parental lines [5]. Both fresh weight of aerial plant parts and rosette leaf expansion are also decreased in mutants of *EDM3* and *IBM2* (Fig 1A and 1B and S1 Fig). We used the previously described *sg1-3* mutant allele of *IBM2* [11], *edm2-2* allele of *EDM2* [13] and *edm3-2* of EDM3 [17] for all experiments in this study. We further found that *edm2*, *edm3*, and *ibm2* mutants have shorter primary roots (Fig 1C and S1 Fig). Moreover, only the longer isoforms of EDM3 and IBM2 we previously described, EDM3L and IBM2L [17], can rescue each of these two growth-related defects in *edm3* or *ibm2* mutants, respectively. Functional complementation by the shorter isoforms EDM3S and IBM2S failed as the respective transgenic lines exhibit similar phenotypes as their parental mutants (S1 Fig). Overall, these data show that EDM3L and IBM2L, like EDM2, positively affect overall plant growth.

### Enhanced basal immunity in *edm2*, *edm3* and *ibm2* suppresses plant growth

To examine if growth defects in *edm2*, *edm3* and *ibm2* mutants are caused by their constitutively enhanced basal immunity, we used Arabidopsis lines with defects in defense responses controlled by salicylic acid (SA). The *sid2-2* mutant is compromised in the defense-associated biosynthesis of this phytohormone [24], while the transgenic *NahG* line cannot accumulate SA, due to the expression of a bacterial SA hydroxylase gene [25]. The *pad4-1* mutant is deficient in a SA-responsive signaling step [26]. We crossed *edm3* and *ibm2* mutants to each of these three lines and performed experiments with lines homozygous for each altered component (mutation or transgene). In addition, we generated *edm2-2;sid2-2* and *edm2-2;pad4-1* double mutants. We first examined if enhanced immunity in *edm2*, *edm3* and *ibm2* is reduced by blockage of SA-mediated defense. We tested Arabidopsis seedlings one week after infection with the Noco2 isolate of the pathogenic oomycete *Hyaloperonospora arabidopsidis* (*Hpa*) infection, which is virulent on wild type Col-0 plants and triggers basal defense in this accession [27]. As shown previously [17], *edm2*, *edm3* and *ibm2* plants show reduced susceptibility against *Hpa*Noco2 compared to Col-0, indicating that basal defense in these single mutants is enhanced (Fig 2A–2D). However, we observed complete loss of this enhanced basal defense phenotype in all double mutants as well as the *edm3-2;NahG* and *sg1-3/ibm2;NahG* lines.

We further observed the growth-related effects on the fresh weight of aerial plant parts and primary root length of *edm2*, *edm3* and *ibm2* mutants to be partly or fully reversed to wild type levels in the respective double mutants with *sid2-2*, or *pad4* (Fig 2E–2G and S2 Fig). We made the same observation with *edm3;NahG* and *ibm2;NahG* lines (Fig 2E–2G and S2 Fig). One deviation are the fresh weight measurements of *ibm2* plants in Fig 2G, which seem lower than Col-0, but are not significantly different from this wild type reference line. The Student's t-test *p* value for this is slightly above the significance cutoff. However, compared to Col-0 we observed fresh weight of aerial plant parts and overall growth of *ibm2* plants to be significantly reduced with very high consistency in all other experiments of this study (Figs 1, 2E, 6B; S1B, S1E & S1F, S2A, S2D & S2E and S3). Still the fresh weight of aerial parts of *ibm2;pad4* double

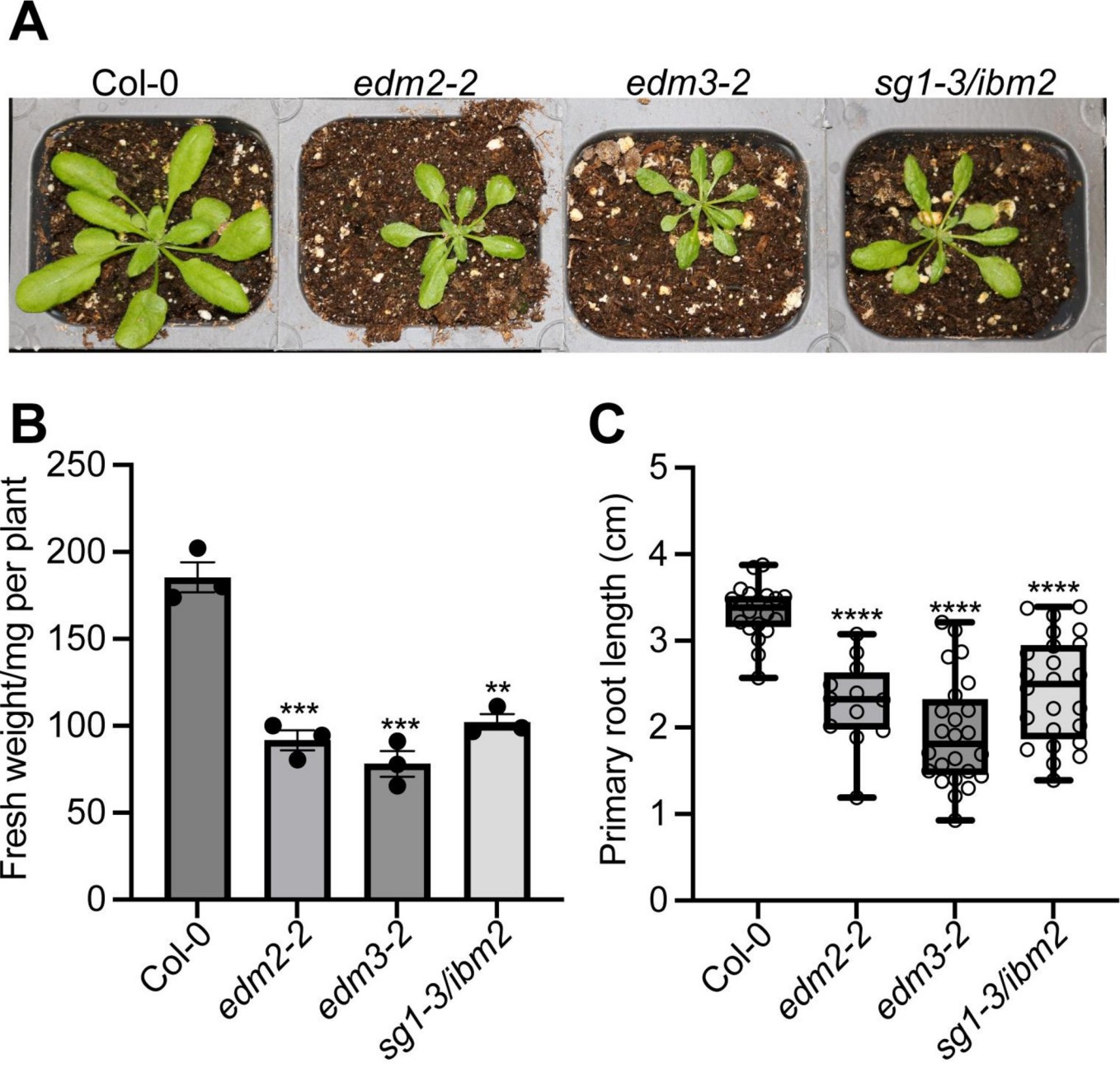

**Fig 1. *EDM2*, *EDM3* and *IBM2* positively affect overall plant growth. A**. Representative images of 25-day-old Arabidopsis plants of the indicated genotypes grown in soil. **B**. Average fresh weight of aerial parts from 25-day-old whole plants of the indicated genotypes. Error bars represent standard errors from three independent experiments, each with eighteen plants. **C**. Primary root length of 7-day-old plants of the indicated genotypes grown on agar plates. Primary root lengths of more than twelve plants of each genotype were measured using ImageJ. Data information: Asterisks indicate significant differences compared to Col-0 based on Student's t-test. (**, $p < 0.01$; ***, $p < 0.001$; ****, $p < 0.0001$).

mutants in this set of experiments is significantly higher than that of the *ibm2* single mutant (Fig 2G).

Overall, these findings show that enhanced immunity in mutants of *EDM2*, *EDM3* and *IBM2* relies on the SA-dependent basal defense pathway. As this pathway is further required

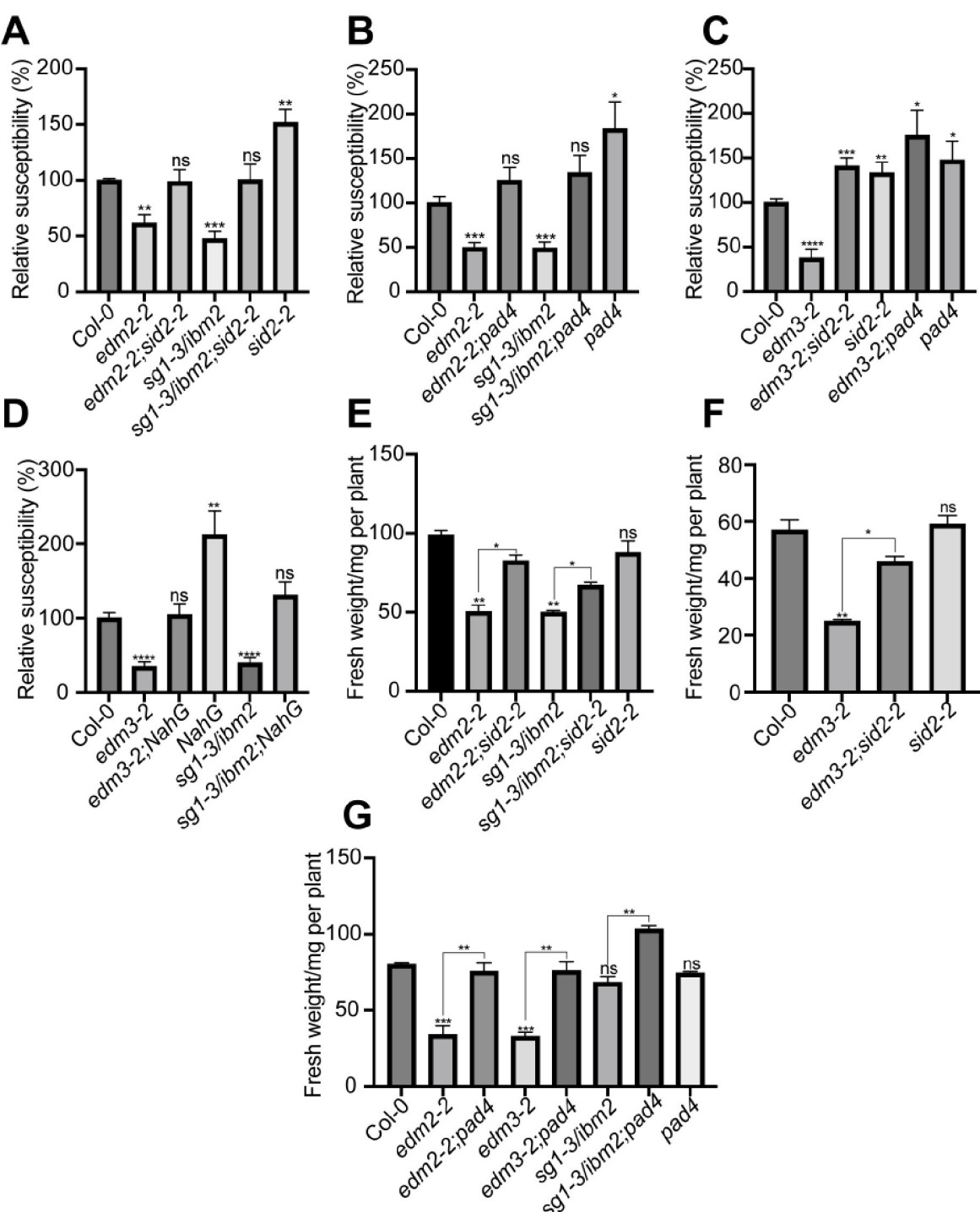

**Fig 2. Enhanced basal immunity in mutants of *EDM2*, *EDM3* and *IBM2* requires salicylic acid, *SID2*, and *PAD4*. A-D**. Enhanced basal immunity in *edm2*, *edm3* and *ibm2* mutants was disrupted by introducing the *sid2-2* or *pad4* mutations or the *NahG* transgene. Two-week-old seedlings were sprayed-inoculated with 3 x 10⁴/ml *Hpa*Noco2 spores. *Hpa*Noco2 spores were counted one week post infection, which were divided by the corresponding fresh weight and shown as percentage relative to wild type. **E-G**. Fresh weight of aerial parts from 24-day-old whole plants (**E**), 23-day-old whole plants (**F**) or 25-day-old whole plants (**G**) of the indicated genotypes, which were grown on soil. Data information: Data shown in each separate panel were generated simultaneously. Error bars represent standard errors from three independent experiments. Asterisks indicate significant differences compared to Col-0 based on Student's t-test (A-D) or by one-way ANOVA with Brown-Forsythe and Welch's test (E) or by ordinary one-way ANOVA (F & G). (*, p < 0.05; **, p < 0.01; ***, p < 0.001; ****, p < 0.0001; ns, no significance).

for the reduction of aerial part fresh weight and primary root length in *edm2*, *edm3* and *ibm2* plants, we conclude that the growth-related effects are a consequence of the enhanced basal defense phenotype of these mutants.

## *EDM2*, *EDM3* and *IBM2* suppress ROS-generating peroxidase activity by down-regulation of type III peroxidase genes

A reactive oxygen species (ROS)-generating oxidative burst is an early event among plant immune responses [19–23, 28–30]. In Arabidopsis, NADPH oxidases and cell wall-associated type III peroxidases are two main sources of ROS in the apoplast [30, 31]. ROS are not only involved in immune response but also affect plant growth. ROS accumulation can stiffen cell walls by cross-linking its components, thereby inhibiting cell expansion [32]. Due to these dual effects, ROS have been linked to the defense-growth trade-off as a shared component of both processes [33].

Re-inspecting our previous RNA-seq analyses on *EDM2*-mediated transcriptome changes [16] we found 27 type III peroxidase genes and two NADPH oxidase genes, *RbohD* and *RbohF*, to be significantly up-regulated in *edm2* plants compared to their wild type parental line, while the gene encoding the $H_2O_2$ scavenging enzyme, *CAT1*, is significantly down-regulated in this mutant. Since over-accumulation of the ROS $H_2O_2$ can reduce cell wall extensibility [32], the joint upregulation of type III peroxidase and NADPH-oxidase genes may contribute to growth-related effects in *edm2* plants.

We selected 11 of these 27 type-III peroxidase genes, which show either particularly strongly increased transcript levels in *edm2* plants or are possible direct target genes of EDM2 and IBM2 based on previous ChIP-seq analyses [18]. For nine of these 11 genes (except *PRX31* and *PRX62*) we consistently observed in qRT-PCRs constitutively elevated transcript levels in the tested *edm2-2*, *edm3-2* and *sg1-3/ibm2* mutants compared to Col-0 (Fig 3A). While we could not confirm the up-regulation of *RbohD* and *RbohF* (Fig 3B), we observed by qRT-PCR a significant up-regulation of *RbohB*, another member of this family of NADPH oxidase genes in all three tested mutants (Fig 3B).

We further tested if peroxidase enzyme activity is also increased in these mutants. Type III peroxidases are known to be localized to the cell wall. Thus, the transcriptional upregulation of these enzymes we observed in mutants of *EDM2*, *EDM3* and *IBM2* should lead to increased peroxidase activity in their ionically bound protein fractions. Indeed, we found that compared to Col-0, the *edm2*, *edm3* and *ibm2* mutants jointly exhibit a noticeable increase in constitutive peroxidase activity in the ionically bound fraction (Fig 3C), but not the soluble fraction (Fig 3D).

To test if the increased peroxidase activity contributes to the growth defects observed in *edm2*, *edm3* and *ibm2* plants, we treated them with 20μM of the peroxidase inhibitor salicylhydroxamic acid (SHAM), at day 14 and day 16 and compared their rosette areas at day 25. The concentration and timing of SHAM treatment we applied has been previously reported as suitable [34]. We observed a significant increase in rosette areas in all tested mutants after SHAM treatment compared to untreated plants, while there is no difference in wild type after treatment with this inhibitor (Fig 4). Overall, these data suggest that increased peroxidase activity contributes to growth defects in *edm2*, *edm3* and *ibm2* mutants.

## Basal *Hpa*Noco2 resistance in mutants of *EDM2*, *EDM3* and *IBM2* is decreased by inhibiting peroxidase activity

To examine if enhanced basal resistance in *edm2*, *edm3* and *ibm2* mutants is partially due to the increased type III peroxidase activity, we pre-treated all three mutants with SHAM 24

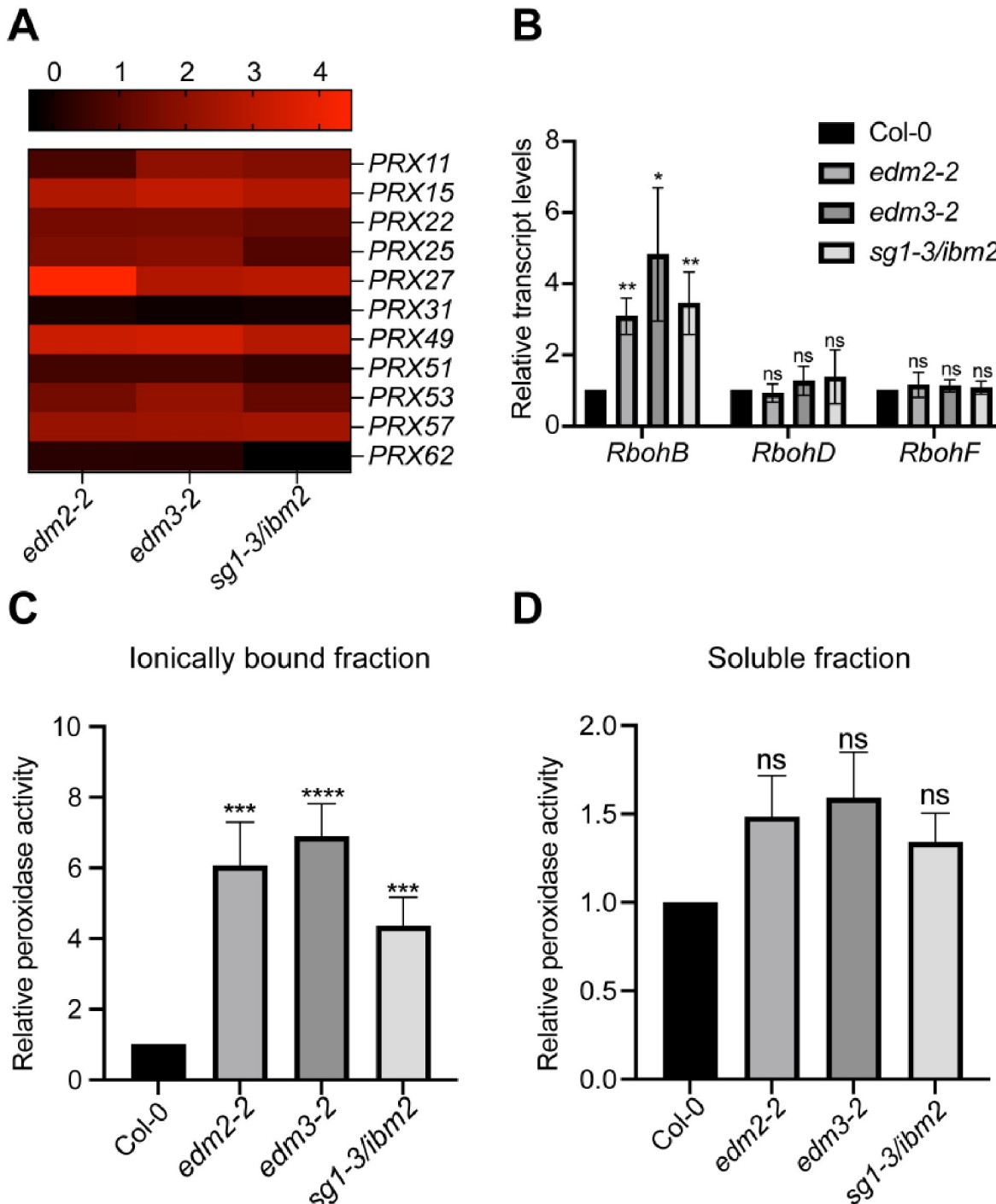

**Fig 3. *EDM2*, *EDM3* and *IBM2* suppress expression levels of genes involved in the generation of ROS as well as activity levels of peroxidases. A**. The transcript levels of selected peroxidase genes in mutants of *EDM2*, *EDM3* and *IBM2* in the aerial parts measured by qRT-PCR. Log2-Fold Change (FD) relative to transcript levels in wild type was used to create this heatmap. The brightest red of the above scale represents the highest increase relative to wild type plants. Numbers above the scale bar represent log2 fold change values of transcript level increases relative to Col-0. **B**. Transcript levels of NADPH oxidase genes in mutants of *EDM2*, *EDM3* and *IBM2* in the aerial parts measured by qRT-PCR. **C & D**. Relative peroxidase activity in mutants of *EDM2*, *EDM3* and *IBM2*. Peroxidase activity was measured in the aerial parts of two-week-old plants, divided by fresh weight and shown relative to the wild type plants. Data information: Error bars represent standard errors from three independent experiments. Asterisks indicate significant differences analyzed by student's t-test. (*, $p < 0.05$; **, $p < 0.01$; ****, $p < 0.0001$; ns, no significance).

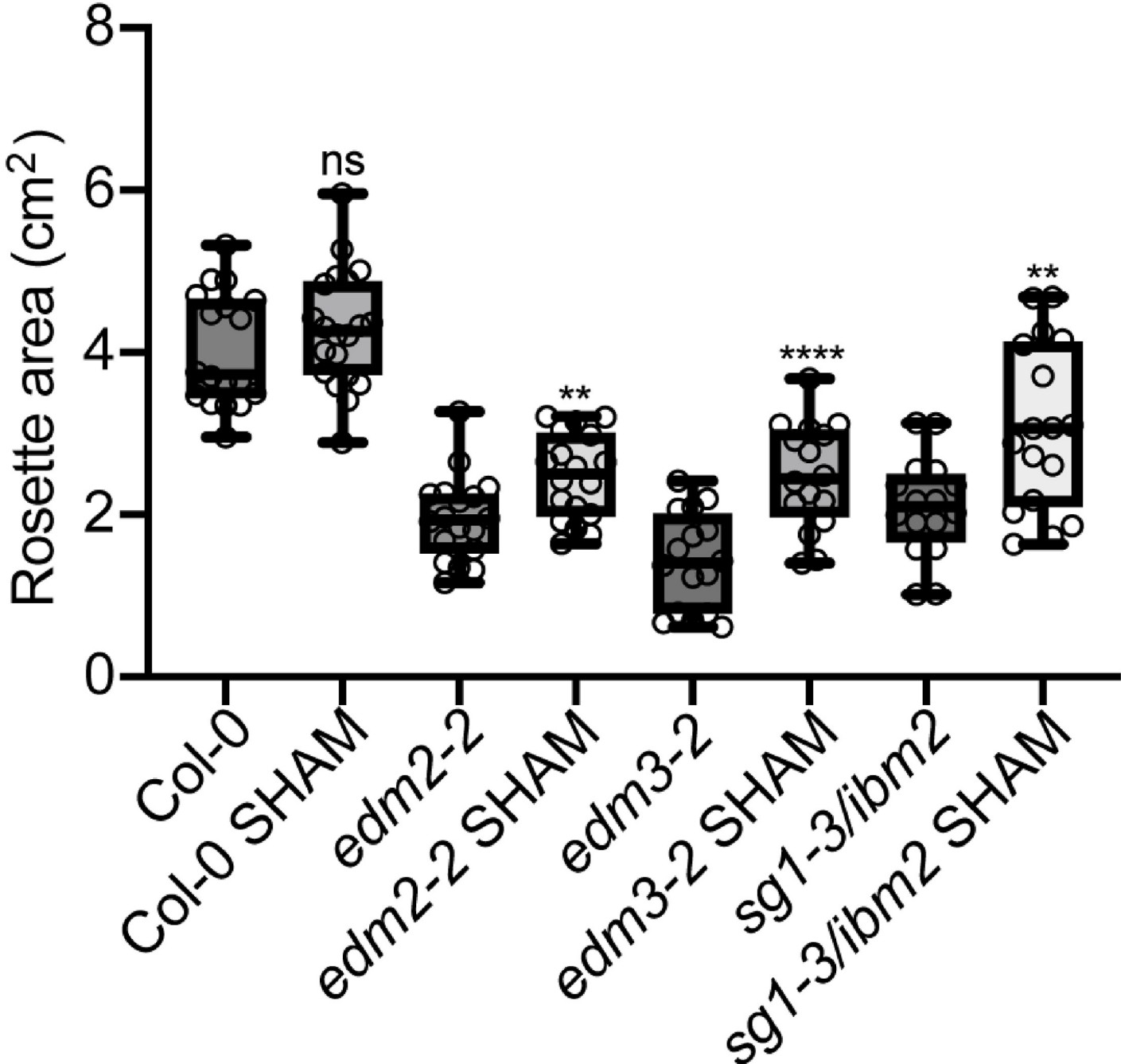

**Fig 4. Growth defects in mutants of EDM2, EDM3 and IBM2 are partly rescued by inhibiting peroxidase activity.** Rosette areas of 25-day-old plants of the indicated genotypes. 20μM SHAM was applied to soil grown plants at day 14 and day 16. Rosette area (n ≥ 16) was measured using ImageJ. Data information: Asterisks indicate significant differences between SHAM and non-SHAM treated plants of the same genotype based on Student's t-tests. (*, $p < 0.05$; **, $p < 0.01$; ****, $p < 0.0001$; ns, no significance).

hours before infection with *Hpa*Noco2. Compared to untreated plants we observed a larger number of *Hpa*Noco2 spores in all three mutant plants after treatment with SHAM and a mild increase in Col-0 wild type plants with SHAM treatment (Fig 5), suggesting increased peroxidase activity contributes to enhanced immunity in all three mutants.

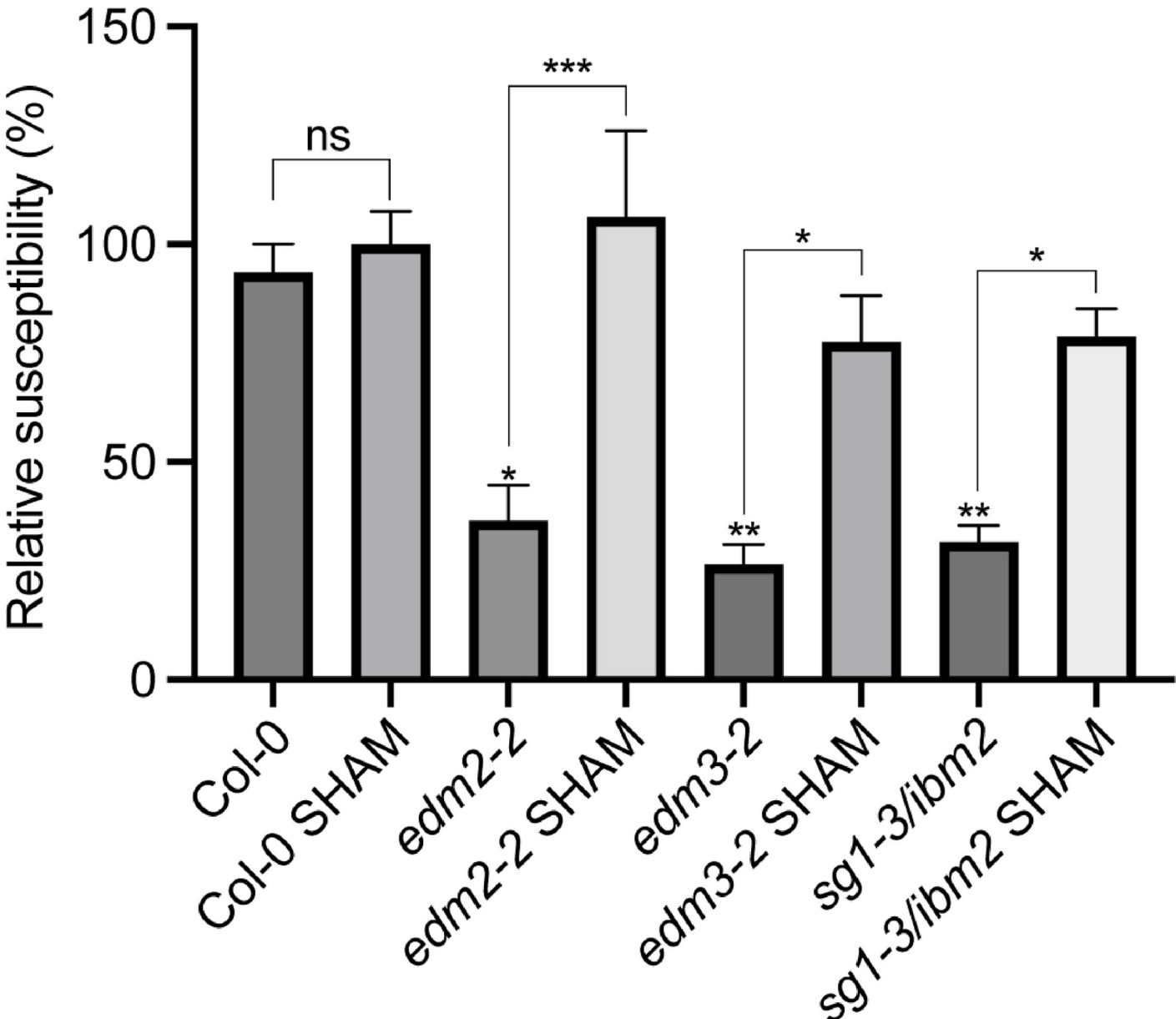

**Fig 5. Increased type III peroxidase activity is likely to contribute to enhanced basal defense in mutants of *EDM2*, *EDM3* and *IBM2*.** Two-week-old plants of the indicated genotypes were pre-treated with 20μM SHAM 24 hours before sprayed-inoculated with 1 x 10$^4$/ml *Hpa*Noco2 spores. *Hpa*Noco2 spores numbers were determined one week after infection. Spore numbers were normalized to the corresponding fresh weights. Data information: Error bars represent standard errors from three independent experiments. Asterisks indicate significant differences between plants of the same genotype based one-way ANOVA. (*, $p < 0.05$; **, $p < 0.01$; ***, $p < 0.001$; ns, no significance).

### IBM1L partly rescues growth defects and disease susceptibility in mutants of *EDM2*, *EDM3* and *IBM2*

At the epigenome and transcriptome levels mutations in the histone H3 demethylase gene *IBM1* phenocopy many effects of *edm2*, *edm3* and *ibm2* mutations. Mutants of *IBM1* are also growth-retarded. *EDM2*, *EDM3* and *IBM2* are known to promote the expression of the functional full-length IBM1 isoform (IBM1L) by suppressing proximal transcript polyadenylation

at an alternative polyadenylation signal in a heterochromatic repeat region in the seventh *IBM1* intron [8, 9]. To examine dependency of *EDM2*, *EDM3* and *IBM2*-mediated basal defense and growth-related effects on *IBM1*, we transformed into *edm2*, *edm3* and *ibm2* mutants a genomic clone containing the entire *IBM1* gene (*gIBM1*) or a mutated version of this clone with a deletion of the heterochromatic repeats in its seventh intron (*gIBM1ΔHR*). While *gIBM1* was shown before to rescue various phenotypes of *ibm1* mutants, *gIBM1ΔHR*, but not *gIBM1*, rescues low expression levels of the functional full-length IBM1L isoform and leaf abnormalities in *edm2* and *ibm2* mutants [8, 9]. The *gIBM1ΔHR* clone lacks the proximal alternative polyadenylation signal in intron 7 and, therefore, does not require *EDM2*, *EDM3* and *IBM2* for proper expression of IBM1L. We observed that *Hpa*Noco2 susceptibility, aerial part fresh weight and primary root length were at least partially restored to wild type levels in *edm2*, *edm3* and *ibm2* mutants containing *gIBM1ΔHR*, but not *gIBM1* (Fig 6A and 6B and S3 Fig). Compared to Col-0 in most cases *Hpa*Noco2 susceptibility levels, aerial part fresh weight and primary root length were still significantly reduced in the mutants containing *gIBM1ΔHR*. However, in all cases these values were also significantly increased compared to their respective *gIBM1*-containing counterparts. Thus, the histone H3 demethylase gene *IBM1*, is at least partially responsible for *EDM2*, *EDM3* and *IBM2*-mediated suppression of basal defense and the resulting promoting effects on growth.

## Discussion

Numerous Arabidopsis mutants with constitutively activated immunity have been described before, such as *cpr (constitutive expresser of PR genes)* [22], *cim (constitutive immunity)* [23], *lsd (lesion-simulating disease resistance)* [20], *acd (accelerated cell death)* [35], *dnd (defense no death)* [36] and *edr (enhanced disease resistance)* [37] mutants. Common to mutants of these classes are constitutively enhanced expression levels of defense-related genes, enhanced basal defense and retarded growth or other developmental defects. These effects are typically dependent on SA-mediated defense signaling. Mutants of *EDM2*, *EDM3* and *IBM2* exhibit the same set of phenotypic effects. Using double mutants we showed enhanced basal defense and retarded rosette and root growth of *edm2*, *edm3* and *ibm2* plants to depend on the SA biosynthesis gene *SID2* and the SA signaling gene *PAD4*. Expression of the bacterial SA hydroxylase gene *NahG* leads to the same effects in *edm3* and *ibm2* plants. Thus, both the enhanced basal defense and retarded growth phenotypes of the *edm2*, *edm3* and *ibm2* mutants are genetically interlinked. As blockage of SA-mediated defense responses rescues the growth defects of these mutants, their growth retardation must be a consequence of the enhanced basal defense. Consequently *EDM2*, *EDM3* and *IBM2* contribute to the defense-growth trade of by limiting the extent of energetically costly basal defense responses, thereby, prioritizing overall growth (Fig 7). We recently showed the coordination of basal defense with the timing of the floral transition to be mediated by the respective longer isoforms of EDM3 and IBM2. This also applies to the growth promoting effects, which is consistent with the view that these processes are interlinked.

While *EDM2*, *EDM3* and *IBM2* suppress basal defense, they have the opposite effect on immunity mediated by the NLR-type immune receptor genes *RPP7* and *RPP4*, the expression of which they promote [4, 12, 14–17]. In particular *RPP7*-mediated immunity may be energetically costly to plants, as this defense mechanism is unusually strong leading to very tight disease resistance [4, 38]. In addition, mutants of *RPP7* exhibit enhanced overall growth, suggesting that even in a resting state this immune receptor consumes energy-related resources (Yan Lai, Tokuji Tsuchiya; Jianqiang. Wang, and Thomas. Eulgem, unpublished). Thus, the contribution to the defense-growth trade-off of *EDM2*, *EDM3* and *IBM2* seems

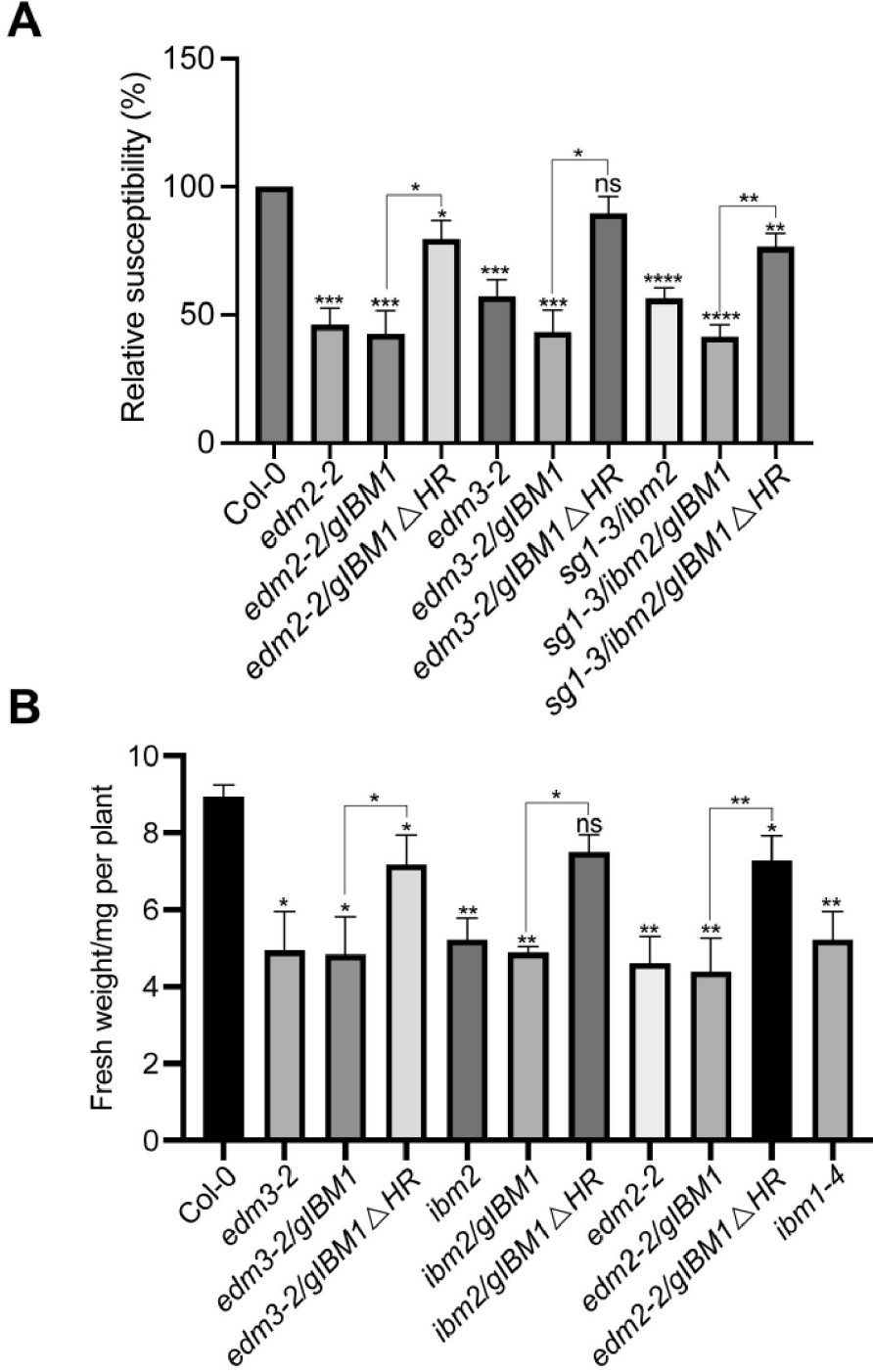

**Fig 6. Growth defects and *Hpa*Noco2 susceptibility were partly rescued by IBM1L. A**. *Hpa*Noco2 susceptibility is rescued in mutants of *EDM2*, *EDM3* and *IBM2* by a genomic *IBM1* clone with a deletion of its heterochromatic repeats in intron 7 (*gIBM1ΔHR*), but not by the complete genomic region lacking the deletion (*gIBM1*). Two-week old plants were spray-inoculated with 3 x $10^4$/ml *Hpa*Noco2 spores. *Hpa*Noco2 spores were counted one week post infection. The resulting spore numbers were divided by the respective fresh weight and shown as percentage relative to wild type. **B**. Fresh weight of aerial parts from twelve-day-old whole plants of the indicated genotypes. Data information: Error bars represent standard errors from three independent experiments. Asterisks indicate significant difference compared to Col-0 based on Student's t-test (*, $p < 0.05$; **, $p < 0.01$; ***, $p < 0.001$; ****, $p < 0.0001$, ns, no significance).

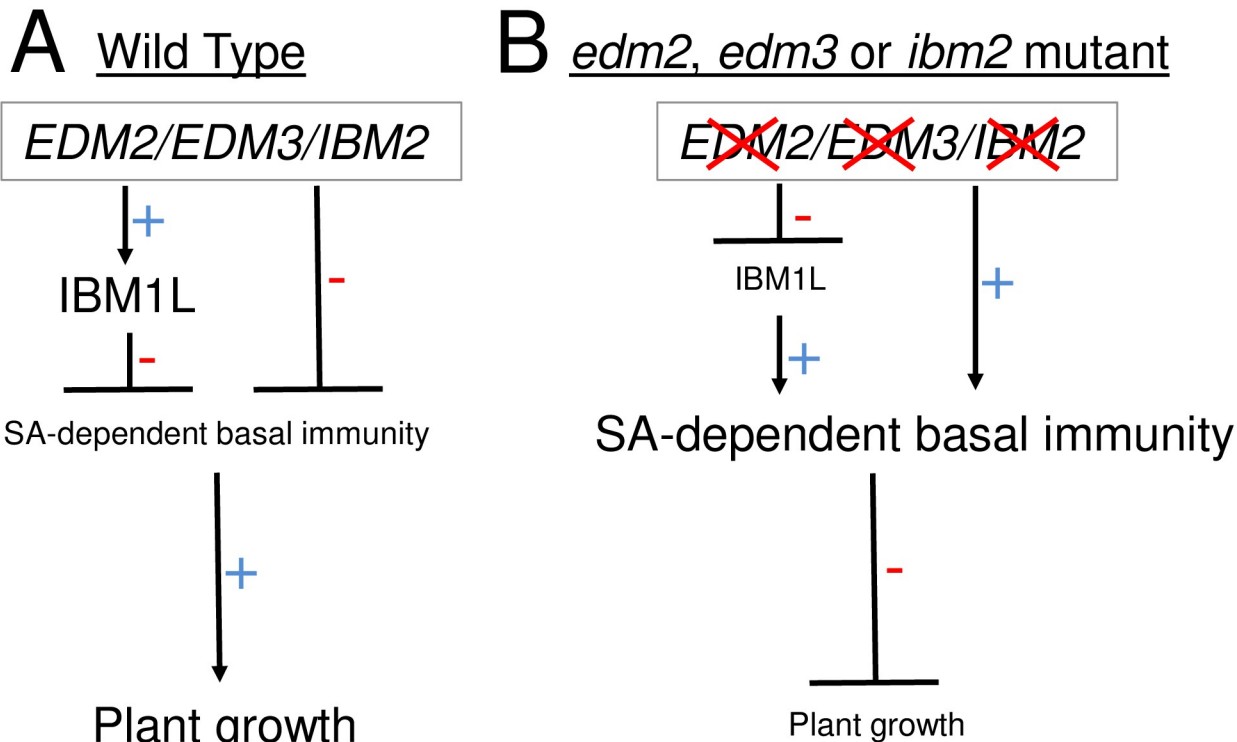

**Fig 7. Model illustrating joint effects of *EDM2*, *EDM3* and *IBM2* on basal immunity and growth. A**. In wild type Arabidopsis, *EDM2*, *EDM3* and *IBM2* suppress SA-dependent basal immunity. This effect is at least partially mediated by their promoting effect on expression of the full-length IBM1 isoform IBM1L. Reduced basal immunity has a promoting effect on plant growth. **B**. In single mutants of *EDM2*, *EDM3* or *IBM2*, SA-dependent basal defense responses are de-repressed resulting in reduced plant growth. The increase if SA-dependent basal immunity is at least partially due to reduced levels of IBM1L.

complex. Rather than serving as general suppressors immune responses, *EDM2*, *EDM3* and *IBM2* seem to mediate a certain balance of defense-related processes, by having counter-directional effects and promoting some defense mechanisms at the expense of others. In addition, they coordinate the extent of basal defense with at least one important developmental transition, the one from vegetative growth to flowering.

As a likely joint contributor of the elevated basal immunity in *edm2*, *edm3* and *ibm2* plants we identified constitutively increased activity of type III peroxidases, a class of enzymes known to contribute to the defense-associated oxidative burst. The oxidative burst is a very early immune response, occurring within minutes when plants are invaded by pathogens [39]. Plant defenses require ROS generated in the oxidative burst, which serve as defensive antimicrobial compounds, cross-linkers of cell wall macromolecules as well as defense signaling molecules [40, 41]. Expression of a set of type III peroxidase-encoding genes is constitutively upregulated in *edm2*, *edm3* and *ibm2* mutants. Consistent with the known localization of type III peroxidases to plant cell walls [31], we observed increased peroxidase activity in anionic protein fractions and not in soluble protein fractions of *edm2*, *edm3* and *ibm2* plants. Blockage of peroxidase activity by the inhibitor SHAM partially reversed the basal defense and growth levels of these mutants to those observed in their wild type parents. Thus, increased type III-mediated accumulation of ROS may contribute to the enhanced basal defense phenotype observed in *edm2*, *edm3* and *ibm2* plants. Apoplastic ROS accumulation may also reduce cell wall extensibility possibly due to lignin formation and cross-linking of other macromolecular

components [32, 42], thereby suppressing plant cell growth. However, using standard $H_2O_2$ detection assays, we were unable to detect any significant increase of this reactive oxygen species in *edm2*, *edm3* and *ibm2* plants. Perhaps the actual increase of ROS is only incremental and too weak to be detectable by the assays we applied.

Many epigenomic and transcriptomic effects caused by *EDM2*, *EDM3*, *IBM2* are known to be mediated by their direct target gene *IBM1*. By suppressing proximal polyadenylation of IBM1 transcripts *EDM2*, *EDM3* and *IBM2* jointly promote expression of the functional full-length IBM1L isoform. Here we found suppression of basal defense as well as the promotion of rosette and root growth by *EDM2*, *EDM3* and *IBM2* to depend at least partially on IBM1L. We already reported overlapping effects on the expression of defense-related and immune receptor genes by *EDM2* and *IBM1* [16]. However, we also observed that each of these regulators affects transcript levels of separate sets of defense-related and immune receptor genes. *IBM1*, in contrast to *EDM2*, *EDM3* and *IBM2*, does not affect *RPP7* expression and immunity-mediated by this NLR gene.

Collectively our results show joint effects of *EDM2*, *EDM3* and *IBM2* on basal immunity and overall plant growth to be genetically interlinked and functionally connected by their dependency on *SID2*, *PAD4* and *IBM1*. This view is further supported by effects we observed using the type III peroxidase inhibitor SHAM. The promotion of plant root and rosette growth by *EDM2*, *EDM3* and *IBM2* must be a consequence of their suppressive effects on basal immunity. Generally these three chromatin-associated regulators seem to contribute to a balanced defense-growth trade off by prioritizing overall plant growth over basal defense responses. Results of this study capture only a fraction of the roles *EDM2*, *EDM3* and *IBM2* have in coordinating immune mechanisms and developmental/growth-related processes. As outlined above, they also align the extent of basal defense with the timing of the floral transition and prioritize *RPP7*-mediated race-specific immunity over basal defense. Additional research is needed to uncover the mechanics of these coordinative processes and to understand the full scope and significance their have in mediating a balanced defense-growth trade off.

## Materials and methods

### Plant material and growth conditions

All the genotypes except specified are in Col-0 background. The following single mutants used in this study: *edm2-2* [4], *edm3-2* [17], *sg1-3/ibm2* [11], *sid2-2* [43], *pad4* [26], and *ibm1-4* [16]. The following double mutants used in this study: *edm2-2;edm3-2*, *edm2-2;sg1-3/ibm2*, *edm3-2; sg1-3/ibm2* were described previously [17]. The following transgenic lines used in this study: *EDM3S[np]-1*, *EDM3S[np]-2*, *EDM3L[np]-1*, *EDM3L[np]-2*, *IBM2S-1* and *IBM2L-1* were described previously [17], *NahG* [44]. The genomic *IBM1* region (*gIBM1*) or *gIBM1* with a deletion of the heterochromatic repeats in its 7th intron (*gIBM1ΔHR*) were cloned as described previously [9] and transformed into *edm2-2*, *edm3-2* and *sg1-3/ibm2*, respectively, to generate the *gIBM1* and *gIBM1ΔHR* transgenic lines. Double mutants were made by crossing and F2 seeds were screened by genotyping PCR to select homozygotes. Plants are grown either on ½ MS medium [45] or in soil under long day (16h/8h light-dark cycles).

### Primary root length measurement

Plants were grown vertically on ½ MS medium for five or seven days. Primary root length of the indicated genotypes was measured using ImageJ.

## Rosette area measurements

Seedlings were untreated or treated with 20μM SHAM (S607-5G, Sigma) at day 14 and day16. Rosette areas were pictured and measured using ImageJ at day 25.

## RNA extraction and qRT-PCR

Total RNA of the aerial parts of two weeks old plants was extracted using Trizol reagent (Life technologies) according to the instructions of the manufacturer. 1μg of RNA was reversed transcribed into cDNA using Maxima Reverse Transcriptase (Fisher scientific). The cDNA products were used for Real-time PCR with CFX CONNECT detection system (Bio-Rad). All primers used are listed in S1 Table.

## Peroxidase activity measurement and inhibitor experiments

Two weeks old seedlings were used for peroxidase activity measurement as described previously [46]. Effects of peroxidase inhibition were tested using salicylhydroxamic acid (SHAM).

## *Hyaloperonospora arabidopsidis* infection assay

*Hpa*Noco2 infection assays were performed as previously described [47]. Briefly, two week-old soil-grown Arabidopsis seedlings were infected by foliar spray with a water-based suspension of 1–3 x $10^4$ spores/ml of asexual *Hpa*Noco2 spores, then covered by saran wrap and kept at 18˚C under short day conditions (8h-day/16h-night cycle) for 7 days. The average numbers of spores per 12 seedlings were determined using a hemocytometer and normalized to the fresh weight of the aerial plant parts.

## Supporting information

**S1 Fig. Only expression longer EDM3 and IBM2 isoforms can rescue growth defects in *edm3* and *ibm2* mutants.** A Fresh weight of aerial parts from 15-day-old Col-0, *edm3-2* or *EDM3* isoform-specific complementation lines (*EDM3S^{np}-1*, *EDM3L^{np}-1)* grown in soil. *EDM3S^{np}-1* and *EDM3L^{np}-1* express in the *edm3-2* mutant background either the short or long EDM3 isoform, respectively, driven by the native *EDM3* promoter (*np*). B. Fresh weight of aerial parts from 12-day old Col-0, *ibm2* and *IBM2* isoform-specific complementation lines (*IBMS-1, IBM2L-1)* grown in soil. *IBMS-1, IBM2L-1* express in the *ibm2* mutant background either the short or long IBM2 isoform, respectively, driven by the native *IBM2* promoter. The *sg1-3* mutant allele of *IBM2* was used for all experiments. C & E. Primary root length of 5-day-old plants of Col-0, *edm3-2* and *ibm2* plants as well as EDM3-and IBM2-isoform specific complementation lines grown on agar plates. D & F. Representative images of plants used in panels C and E. Data information: Error bars represent standard errors from three independent experiments. Asterisks indicate significant differences compared to Col-0 based on Student's t-test. (*, $p < 0.05$; **, $p < 0.01$; ****, $p < 0.0001$; ns, no significance). $n \geq 28$ (C) and $n \geq 20$ (E).
(TIF)

**S2 Fig. Primary root length of 5-day-old plants of the indicated genotypes.** Primary root length of the indicated genotypes was measured using ImageJ. Data information: Data shown in each separate panel were generated simultaneously. Asterisks indicate significant differences compared to Col-0 based on one-way ANOVA (A-E). (*, $p < 0.05$; **, $p < 0.01$; ***, $p < 0.001$; ****, $p < 0.0001$; ns, no significance). $n \geq 18$ (A), $n \geq 24$ (B), $n \geq 33$ (C), $n \geq 17$ (D), $n \geq 21$

(E).
(TIF)

**S3 Fig. Short primary root length in mutants of *EDM2*, *EDM3* and *IBM2* was partly rescued by wild type expression of *IBM1L*.** Primary root lengths of five day-old plants of the indicated genotypes were measured using ImageJ. Error bars represent standard errors. Asterisks indicate significant differences compared to Col-0 based on one-way ANOVA. (\*\*, $p < 0.01$; \*\*\*\*, $p < 0.0001$). $n \geq 20$.
(TIF)

**S1 Table. Primers used in this study.**
(DOCX)

## Acknowledgments

We thank Drs. Xuemei Chen and Meng Chen (both University of California, Riverside, USA) for advice on this project.

## Author Contributions

**Conceptualization:** Jianqiang Wang, Thomas Eulgem.

**Funding acquisition:** Thomas Eulgem.

**Investigation:** Jianqiang Wang.

**Supervision:** Thomas Eulgem.

**Writing – original draft:** Jianqiang Wang.

**Writing – review & editing:** Jianqiang Wang, Thomas Eulgem.

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
