## [Decision Letter · Decision Letter 0]

28 Sep 2023

PONE-D-23-28621Growth deficiency and enhanced basal Immunity in Arabidopsis thaliana mutants of EDM2, EDM3 and IBM2 are genetically interlinkedPLOS ONE

Dear Dr. Eulgem,

Thank you for submitting your manuscript to PLOS ONE. After careful consideration, we feel that it has merit but does not fully meet PLOS ONE’s publication criteria as it currently stands. Therefore, we invite you to submit a revised version of the manuscript that addresses the points raised during the review process.

We look forward to receiving your revised manuscript.

Kind regards,

Anwar Hussain

Academic Editor

PLOS ONE

[This work was supported by National Science Foundation (NSF) grant IOS-1457329 to TE.]

 [This work was supported by National Science Foundation (NSF) grant IOS-1457329 to TE. 

URL of funders web site: https://www.nsf.gov/div/index.jsp?div=IOS

The funders had no role in study design, data collection and analysis, decision to publish, or preparation of the manuscript.]

4. We note that Figure 1A, S1D and S1F in your submission contain copyrighted images. All PLOS content is published under the Creative Commons Attribution License (CC BY 4.0), which means that the manuscript, images, and Supporting Information files will be freely available online, and any third party is permitted to access, download, copy, distribute, and use these materials in any way, even commercially, with proper attribution. For more information, see our copyright guidelines: http://journals.plos.org/plosone/s/licenses-and-copyright.

A. You may seek permission from the original copyright holder of Figure 1A, S1D and S1F to publish the content specifically under the CC BY 4.0 license. 

B. If you are unable to obtain permission from the original copyright holder to publish these figures under the CC BY 4.0 license or if the copyright holder’s requirements are incompatible with the CC BY 4.0 license, please either i) remove the figure or ii) supply a replacement figure that complies with the CC BY 4.0 license. Please check copyright information on all replacement figures and update the figure caption with source information. If applicable, please specify in the figure caption text when a figure is similar but not identical to the original image and is therefore for illustrative purposes only.

5.Please include a copy of Table 4.1 which you refer to in your text on page 17.

6. We notice that your supplementary figures are uploaded with the file type 'Figure'. Please amend the file type to 'Supporting Information'. Please ensure that each Supporting Information file has a legend listed in the manuscript after the references list.

Reviewers' comments:

Reviewer's Responses to Questions

**Comments to the Author**

1. Is the manuscript technically sound, and do the data support the conclusions?

Reviewer #1: Yes

Reviewer #2: Yes

2. Has the statistical analysis been performed appropriately and rigorously? 

Reviewer #1: Yes

Reviewer #2: Yes

3. Have the authors made all data underlying the findings in their manuscript fully available?

Reviewer #1: Yes

Reviewer #2: Yes

4. Is the manuscript presented in an intelligible fashion and written in standard English?

Reviewer #1: Yes

Reviewer #2: Yes

5. Review Comments to the Author

Reviewer #1: Introduction

1. Page 3, para 2, line(s) 10 of the page:

Replace “....... are know to jointly affect........” by ...... are known to jointly affect........

2. Page 3, line(s) 13-15 of the page:

Properly punctuate the sentence and remove the word different as given below:

They also work together in an unspecific immune response, effective against a wide range of pathogens, termed basal immunity or basal resistance [16,17].

3. Page 3, line 16 of the page and Page 4, line 11 of the page:

Write “basal defense response or basal defense system” instead of mere writing “basal defense”.

4. Page 4, lines 11-13 of the page:

Rephrase the sentence as: Regarding their role in the suppression of basal defense response and the promotion of the floral transition, only the longer isoforms (EDM3L and IBM2L) are involved and not the shorter ones i.e. EDM3S and IBM2S [17].

5. Page 4, lines 14-16 of the page:

These lines, with use of phrase “we observed” give an impression of results of this (under review paper) which is not the case. It will be better to replace this phrase by “it has been observed” and written as:

Consistent with their role in suppressing the basal defense response, constitutive up-regulation in the transcripts of large overlapping sets of defense-related genes and immune receptor genes was observed in the mutants of EDM2, EDM3 and IBM2 [15–17].

6. Page 5, lines 1-6 of the page:

The last sentence of the introduction is so long (6 lines) and confusing probably because of poor punctuation and articulation that the readers are lost. Rephrase it for clarity.

7. Introduction has no clear hypothesis/question and/or rationale/significance which needs to be added.

RESULTS

8. Page 6, lines 8-10 of the page:

The sentence is not properly structured and is not appropriately punctuated/articulated and hence fails to carry a clear meaning/message. Write it in clear way.

9. Page 8, line 23 of the page:

The reference cite may be written as s [19-23, 28-30] instead of [28][19–23][29,30].

Discussion

10. The authors need to provide a model for conclusive mechanism that relates the basal immunity response to growth response in these mutants.

Materials and Methods

11. Page 16, line 21 of the page:

Provide reference for MS medium (Murashige and Skoog, 1962) given below:

Murashige T, Skoog F: A revised medium for rapid growth and bioassays with tobacco tissue culture. Physiologia Plantarum 1962, 15:473-497.

12. Page 23, line 9 of the page:

S1 Table gives the sequences of the primers used in this study. The authors provided reference for only one set/pair of primers. It is advised to provide reference(s) for all of these primers.

General

13. Problems related to proper punctuation were noticed. A thorough read and subsequent corrections is recommended for this purpose.

Reviewer #2: General comments

Overall, the research paper presents a thorough investigation into the genetic interplay between EDM2, EDM3, and IBM2 genes in Arabidopsis thaliana mutants. The study effectively addresses the defense-growth trade-off, a crucial aspect of plant immunity and development. The methodology is well-designed, incorporating a range of experimental techniques to explore genetic interactions and phenotypic outcomes. Overall, the study is well designed and executed. The following minor comments/suggestions should be addressed.

Stick to the Journal format of paper writing including sequence in the sections and references style

Specific comments

Abstract

Since you mention "We recently reported...", it would be helpful to briefly mention any key findings from your previous work to provide context for this study.

End the abstract with a clear statement summarizing the main takeaway or contribution of the research.

Introduction

When introducing terms like "NLR-type immune receptor" or "cytosine methylation," consider providing brief explanations or references for readers who might not be familiar with these concepts.

It might be helpful to clarify the full names of the genes (EDM2, EDM3, and IBM2) upon their first mention, even though you provide explanations later. This can enhance initial understanding for readers.

While it's implied, consider explicitly stating the hypothesis or research question that your study aims to address. Additionally, mention the specific objectives or goals of your research at the end of the introduction.

Results

The use of specific mutant alleles (sg1-3, edm2-2, edm3-2) is well-described. Providing these details is important for reproducibility and allows other researchers to understand the specific genetic context of your experiments.

The observation that edm2, edm3, and ibm2 mutants have shorter primary roots is a significant finding. when discussing the selection of type-III peroxidase genes, it might be helpful to briefly explain why PRX31 and PRX62 were not included. It would be interesting to speculate on the potential molecular mechanisms underlying this phenotype.

Treatment Timing:

The choice of treatment timing (day 14 and day 16) seems justified, but it would be helpful to provide a rationale for why these specific time points were chosen. This could be based on the growth stage or a previous study's findings.

Dosage of Inhibitor:

The concentration of the peroxidase inhibitor (20μM of SHAM) is specified. It would be beneficial to briefly explain why this particular concentration was chosen, especially if it aligns with prior research or known effective concentrations.

Measurement of Basal Resistance

Clarify how basal resistance is measured. This could involve details about the quantification or assessment of HpaNoco2 spores, including the method used and any potential controls or replicates.

Rationale for Genomic Clones in IBM1L partly rescues growth defects and disease susceptibility in mutants of EDM2, EDM3 and IBM2:

It would be helpful to provide a brief explanation of why these specific genomic clones were chosen and how they relate to the research question.

6. PLOS authors have the option to publish the peer review history of their article (what does this mean?). If published, this will include your full peer review and any attached files.

Reviewer #1: No

Reviewer #2: **Yes: **Muhammad Qadir

---

## [Author Response · Author response to Decision Letter 0]

12 Nov 2023

Dear Editors and Reviewers,

We are grateful for the opportunity to revise our manuscript and for the thoughtful and constructive critique provided by the editor and reviewers. We believe we sufficiently addressed all critique points and the paper is now very much improved. Our responses to the individual points are provided below typed in “blue”. We are submitting a revised manuscript text file with marked track changes as well as a revised clean version. We hope you manuscript can now be accepted for publication in PLOS ONE. 

Thank you and best regards,

Thomas Eulgem

Journal requirements

Response: We reformatted the manuscript text according to the PLOS ONE style templates.

Response: We did this and made corrections. The information is now correct and consistent in both sections.

[This work was supported by National Science Foundation (NSF) grant IOS-1457329 to TE.]

 [This work was supported by National Science Foundation (NSF) grant IOS-1457329 to TE. 

URL of funders web site: https://www.nsf.gov/div/index.jsp?div=IOS. The funders had no role in study design, data collection and analysis, decision to publish, or preparation of the manuscript.]

Response: We removed the statement regarding funding from the Acknowledgement section and added some new text thanking two colleagues for their input. The information we provided in the “funding statement section” is correct and does not need to be altered. 

4. We note that Figure 1A, S1D and S1F in your submission contain copyrighted images. 

Response: Figure 1A, S1D and S1F in our submission do not contain copyrighted images. We took the photos shown there by ourselves and have never published them elsewhere, except in the preprint we submitted via PLOS ONE to bioRxiv. Does bioRxiv now possess the copyrights for these images? If so, why only the photos and not all figures in this preprint, which are identical to the figures in our PLOS ONE submission. Can you please look into this and let us know if a mistake has been made? 

In response to an email explaining the above issue, which I sent to Dr. Hussain on 10/5/23, I received on 10/11/23 an email from Jovencio Noel stating the following “Thank you for letting us know that the mention figures were not copyrighted. Please disregard that instruction and please include your explanation to your cover letter.” Based on this we consider this issue to be resolved. 

5.Please include a copy of Table 4.1 which you refer to in your text on page 17.

Response: “Table 4.1” is a typo. Thank you for catching this. We meant to refer to “S1 Table”, which was submitted along with the original version of this paper. We will submit it also along with the revised version. 

6. We notice that your supplementary figures are uploaded with the file type 'Figure'. Please amend the file type to 'Supporting Information'. Please ensure that each Supporting Information file has a legend listed in the manuscript after the references list.

Response: Thank you and sorry for the mistake. We will make sure to do this correct for the revised version. 

Response: We added two papers to our reference list. Please see below Reviewer 1: point 11 and 

Reviewer 2: points 9 and 10.

Reviewer's Comments to the Author

Reviewer #1: 

INTRODUCTION:

1. Page 3, para 2, line(s) 10 of the page: Replace “....... are know to jointly affect........” by ...... are known to jointly affect........

Response: Done.

2. Page 3, line(s) 13-15 of the page: Properly punctuate the sentence and remove the word different as given below:

They also work together in an unspecific immune response, effective against a wide range of pathogens, termed basal immunity or basal resistance [16,17].

Response: Done.

3. Page 3, line 16 of the page and Page 4, line 11 of the page:

Write “basal defense response or basal defense system” instead of mere writing “basal defense”.

Response: Done.

4. Page 4, lines 11-13 of the page:

Rephrase the sentence as: Regarding their role in the suppression of basal defense response and the promotion of the floral transition, only the longer isoforms (EDM3L and IBM2L) are involved and not the shorter ones i.e. EDM3S and IBM2S [17].

Response: Done. We also replaced “basal defense response”, by “basal immunity” to avoid repetition in the next sentence. 

5. Page 4, lines 14-16 of the page:

These lines, with use of phrase “we observed” give an impression of results of this (under review paper) which is not the case. It will be better to replace this phrase by “it has been observed” and written as:

Consistent with their role in suppressing the basal defense response, constitutive up-regulation in the transcripts of large overlapping sets of defense-related genes and immune receptor genes was observed in the mutants of EDM2, EDM3 and IBM2 [15–17].

Response: Done.

6. Page 5, lines 1-6 of the page:

The last sentence of the introduction is so long (6 lines) and confusing probably because of poor punctuation and articulation that the readers are lost. Rephrase it for clarity.

Response: Done.

7. Introduction has no clear hypothesis/question and/or rationale/significance which needs to be added.

Response: thank you for pointing this out. We added the sentence 

”However, it has been unclear, weather these effects are independent or causally connected” 

after the following statement on the bottom of page 4 

“Thus, edm2, edm3 and ibm2 mutants exhibit three typical effects common to many other Arabidopsis mutants with constitutively elevated levels of basal immunity [19–23]: (1) constitutively elevated expression of defense genes, (2) enhanced basal defense and (3) retarded growth.” 

This way a central question, which we pursued in this study, is added. 

RESULTS:

8. Page 6, lines 8-10 of the page: The sentence is not properly structured and is not appropriately punctuated/articulated and hence fails to carry a clear meaning/message. Write it in clear way.

Response: We rephrased this sentence by starting with “Both”, which should make the meaning of the sentence clearer. The corrected sentence is: 

“Both fresh weight of aerial plant parts and rosette leaf expansion are also decreased in mutants of EDM3 and IBM2”

In addition we made a factual correction. We incorrectly stated in the original version of this paper that we measured whole plant fresh weight, while we actually measured fresh weight of aerial plant parts. This also applies to the other experiments in this paper where fresh weight was measured and was due to miscommunication between the two authors. The conclusions we drew from these experiments do not change. We apologize for this unnecessary error and corrected this throughout the rest of the paper. Please see legend of Fig 1B, last paragraph on page 7, 1st two paragraphs on page 8, legend of Fig. 2E-G, last paragraph on page 12, first paragraph on page 13, legend of Fig 6B as well as legend of S1 Fig A& B. 

9. Page 8, line 23 of the page:

The reference cite may be written as s [19-23, 28-30] instead of [28][19–23][29,30].

Response: Done.

DISCUSSION:

10. The authors need to provide a model for conclusive mechanism that relates the basal immunity response to growth response in these mutants.

Response: Thank you for pointing this out. We added Figure 7, which illustrates the model we developed in this study. 

MATERIALS AND METHODS:

11. Page 16, line 21 of the page:

Provide reference for MS medium (Murashige and Skoog, 1962) given below:

Murashige T, Skoog F: A revised medium for rapid growth and bioassays with tobacco tissue culture. Physiologia Plantarum 1962, 15:473-497.

Response: We added the reference for MS medium.

12. Page 23, line 9 of the page:

S1 Table gives the sequences of the primers used in this study. The authors provided reference for only one set/pair of primers. It is advised to provide reference(s) for all of these primers.

Response: We added the references for the primers previously used to S1 Table. All remaining ones were designed for this study and used here for the first time. 

GENERAL:

13. Problems related to proper punctuation were noticed. A thorough read and subsequent corrections is recommended for this purpose.

Response: Done.

Reviewer #2: 

GENERAL COMMENTS:

Overall, the research paper presents a thorough investigation into the genetic interplay between EDM2, EDM3, and IBM2 genes in Arabidopsis thaliana mutants. The study effectively addresses the defense-growth trade-off, a crucial aspect of plant immunity and development. The methodology is well-designed, incorporating a range of experimental techniques to explore genetic interactions and phenotypic outcomes. Overall, the study is well designed and executed. The following minor comments/suggestions should be addressed.

1. Stick to the Journal format of paper writing including sequence in the sections and references style.

Response: We double checked the manuscript and made corrections where necessary. 

SPECIFIC COMMENTS

Abstract:

2. Since you mention "We recently reported...", it would be helpful to briefly mention any key findings from your previous work to provide context for this study.

Response: Thanks for pointing this out. We changed: 

“We recently reported that EDM2, EDM3 and IBM2 coordinate the extent of basal immunity with the timing of the floral transition.” 

to 

“We recently reported that EDM2, EDM3 and IBM2 coordinate basal immunity with the timing of the floral transition by gradually reducing the extent of this defense mechanism prior to flowering”. 

The new sentence provides a bit more information without distracting from the main focus of the submitted study. 

3. End the abstract with a clear statement summarizing the main takeaway or contribution of the research.

Response: Thank you for this suggestion as well. We added the following sentence to the end of the abstract: 

“Our new results show that repression of basal immunity by EDM2, EDM3 and IBM2 limits negative impact on growth and development.”

Introduction:

4. When introducing terms like "NLR-type immune receptor" or "cytosine methylation," consider providing brief explanations or references for readers who might not be familiar with these concepts.

Response: We had provided three references for the term “NLR” and modified the sentence on “cytosine methylation” to make it clearer for the non-specialist. The latter sentence also has several citations providing background information on “cytosine methylation”. However, we do not want to elaborate more on these concepts in order for the paper to remain focused without distracting side information that is not needed. 

5. It might be helpful to clarify the full names of the genes (EDM2, EDM3, and IBM2) upon their first mention, even though you provide explanations later. This can enhance initial understanding for readers.

Response: Done.

6. While it's implied, consider explicitly stating the hypothesis or research question that your study aims to address. Additionally, mention the specific objectives or goals of your research at the end of the introduction.

Response: Done. Please see point 7 of reviewer 1. 

Results:

7. The use of specific mutant alleles (sg1-3, edm2-2, edm3-2) is well-described. Providing these details is important for reproducibility and allows other researchers to understand the specific genetic context of your experiments.

Response: References for these specific mutants have been provided under “Plant material and growth conditions” in the Material and Methods section. 

8. The observation that edm2, edm3, and ibm2 mutants have shorter primary roots is a significant finding. when discussing the selection of type-III peroxidase genes, it might be helpful to briefly explain why PRX31 and PRX62 were not included. It would be interesting to speculate on the potential molecular mechanisms underlying this phenotype.

Response: We speculated on the molecular mechanism in the discussion section and wrote:

“Thus, increased type III-mediated accumulation of ROS may contribute to the enhanced basal defense phenotype observed in edm2, edm3 and ibm2 plants. Apoplastic ROS accumulation may also reduce cell wall extensibility possibly due to lignin formation and cross-linking of other macromolecular components [33,42], thereby suppressing plant cell growth.” 

As both PRX31 and PRX62 are poorly characterized, we did not include them in this study. 

9. Treatment: Timing: The choice of treatment timing (day 14 and day 16) seems justified, but it would be helpful to provide a rationale for why these specific time points were chosen. This could be based on the growth stage or a previous study's findings.

Response: We did this assay based on a previous study’s findings

Lu, D., Wang, T., Persson, S., Mueller-Roeber, B. and Schippers, J.H., 2014. Transcriptional control of ROS homeostasis by KUODA1 regulates cell expansion during leaf development. Nature communications, 5(1), p.3767. We are referencing now this source in the results section on SHAM treatment and listed this paper in the reference list.

10. Dosage of Inhibitor: The concentration of the peroxidase inhibitor (20μM of SHAM) is specified. It would be beneficial to briefly explain why this particular concentration was chosen, especially if it aligns with prior research or known effective concentrations.

Response: We used this concentration as it was reported previously.

Lu, D., Wang, T., Persson, S., Mueller-Roeber, B. and Schippers, J.H., 2014. Transcriptional control of ROS homeostasis by KUODA1 regulates cell expansion during leaf development. Nature communications, 5(1), p.3767. We are referencing now this source in the results section on SHAM treatment and listes this paper in the reference list. 

11. Measurement of Basal Resistance: Clarify how basal resistance is measured. This could involve details about the quantification or assessment of HpaNoco2 spores, including the method used and any potential controls or replicates.

Response: The details of the quantification of HpaNoco2 spores are now included in the material and methods.

12. Rationale for Genomic Clones in IBM1L partly rescues growth defects and disease susceptibility in mutants of EDM2, EDM3 and IBM2:

It would be helpful to provide a brief explanation of why these specific genomic clones were chosen and how they relate to the research question.

Response: We provided an extensive explanation for this on page 12 at the end of the results section.

---

## [Decision Letter · Decision Letter 1]

16 Jan 2024

Growth deficiency and enhanced basal Immunity in Arabidopsis thaliana mutants of EDM2, EDM3 and IBM2 are genetically interlinked

PONE-D-23-28621R1

Dear Dr. Eulgem

We’re pleased to inform you that your manuscript has been judged scientifically suitable for publication and will be formally accepted for publication once it meets all outstanding technical requirements.

Kind regards,

Anwar Hussain

Academic Editor

PLOS ONE

Additional Editor Comments (optional):

Reviewers' comments:

Reviewer's Responses to Questions

**Comments to the Author**

1. If the authors have adequately addressed your comments raised in a previous round of review and you feel that this manuscript is now acceptable for publication, you may indicate that here to bypass the “Comments to the Author” section, enter your conflict of interest statement in the “Confidential to Editor” section, and submit your "Accept" recommendation.

Reviewer #1: All comments have been addressed

Reviewer #2: All comments have been addressed

2. Is the manuscript technically sound, and do the data support the conclusions?

Reviewer #1: Yes

Reviewer #2: Yes

3. Has the statistical analysis been performed appropriately and rigorously? 

Reviewer #1: Yes

Reviewer #2: Yes

4. Have the authors made all data underlying the findings in their manuscript fully available?

Reviewer #1: Yes

Reviewer #2: Yes

5. Is the manuscript presented in an intelligible fashion and written in standard English?

Reviewer #1: Yes

Reviewer #2: Yes

6. Review Comments to the Author

Reviewer #1: (No Response)

Reviewer #2: (No Response)

7. PLOS authors have the option to publish the peer review history of their article (what does this mean?). If published, this will include your full peer review and any attached files.

Reviewer #1: No

Reviewer #2: **Yes: **Muhammad Qadir

---

## [Editor Report · Acceptance letter]

29 Jan 2024

PONE-D-23-28621R1 

PLOS ONE

Dear Dr. Eulgem, 

I'm pleased to inform you that your manuscript has been deemed suitable for publication in PLOS ONE. Congratulations! Your manuscript is now being handed over to our production team.

Kind regards, 

on behalf of

Dr. Anwar Hussain 

Academic Editor

PLOS ONE